# HFpEF without elevated right ventricular systolic pressure is a favorable prognostic indicator in patients with IPF requiring hospitalization for heart failure

Ryo Yamazaki[1,2], Osamu Nishiyama[1]*, Kazuya Yoshikawa[1], Sho Saeki[1], Hiroyuki Sano[1], Takashi Iwanaga[1], Yuji Tohda[1]

1 Department of Respiratory Medicine and Allergology, Kindai University Faculty of Medicine, Osakasayama, Osaka, Japan, 2 Department of Respiratory Medicine and Allergology, Kindai University Nara Hospital, Ikoma, Nara, Japan

* nishiyama_o@yahoo.co.jp

**Data Availability Statement:** All relevant data are within the manuscript and its Supporting Information files.

## Abstract

### Background

Some patients with idiopathic pulmonary fibrosis (IPF) must be hospitalized because of heart failure (HF), including HF with preserved ejection fraction (HFpEF) and HF with reduced EF (HFrEF). The association between IPF and HF has not been clarified. We retrospectively investigated the clinical features and outcomes of patients with IPF who required nonelective hospitalization because of HF.

### Methods

We examined data from IPF patients who required nonelective hospitalization for HF at the Kindai University Hospital from January 2008 to December 2018. We divided the patients into 3 groups: those with HFpEF without elevated right ventricular systolic pressure (RVSP), those with HFpEF and elevated RVSP, and those with HFrEF. The recurrence rates of HF after discharge and the 30- and 90-day mortality rates of the patients were evaluated.

### Results

During the study period, 37 patients with IPF required hospitalization because of HF. Among the 34 patients included in the study, 17 (50.0%) were diagnosed with HFpEF without elevated RVSP, 11 (32.3%) with HFpEF and elevated RVSP, and 6 (17.6%) with HFrEF. Patients with HFrEF had significantly higher values for B-type natriuretic peptide (BNP) and left ventricular (LV) end-systolic and end-diastolic diameters than patients with the 2 types of HFpEF (BNP: $P = 0.01$ and $P = 0.0004$, LV end-systolic diameter: $P < 0.0001$ and $P < 0.0001$, and LV end-diastolic diameter: $P = 0.01$ and $P = 0.0004$, respectively). Notably, the difference between the LVEFs of the patients with 2 types of HFpEF was not significant. The patients with HFpEF without elevated RVSP had the lowest 30- and 90-day mortality rates (0%, $P = 0.02$ and 11.7%, $P = 0.11$, respectively).

**Funding:** The authors received no specific funding for this work.

**Competing interests:** The authors have declared that no competing interests exist.

## Conclusions

Among patients with IPF, HFpEF without elevated RVSP was the most common type of HF that required hospitalization. Patients with HFpEF without elevated RVSP survived longer than the patients with the other 2 types of HF.

## Introduction

Idiopathic pulmonary fibrosis (IPF) is a specific form of chronic, progressive and fibrosing lung disease of unknown etiology and has a poor prognosis [1]. The natural course of IPF is highly variable [2]. Many patients with IPF require hospitalization related to respiratory and/ or cardiovascular disease [3, 4].

Heart failure (HF) is an important health problem worldwide. The prevalence of HF is estimated to be 1%–2% of the adult population in developed countries [5]. About 50% of patients with HF have a normal or nearly normal left ventricular ejection fraction (LVEF) [6–8]. The prevalence of this type of HF, termed HF with preserved ejection fraction (HFpEF), is increased in older populations [9]. Because many patients with IPF are elderly, HFpEF may affect a high proportion of IPF patients with HF. Additionally, nonelective hospitalization, which is sometimes required for patients with IPF, has recently been recognized as an important clinical outcome [10, 11]. HF can account for some of the nonelective hospitalizations of patients with IPF.

The clinical features and outcomes of patients with IPF and HF, including the HF subgroups of HFpEF and HF with reduced EF (HFrEF), are unknown. This study aimed to clarify the details of the clinical features and outcomes of HF in patients with IPF who required hospitalization.

## Methods

### Patients

We retrospectively reviewed the clinical records of all patients with IPF who required nonelective hospitalization for HF at the Kindai University Hospital from January 2008 to December 2018. The diagnosis of IPF was based on the eligibility criteria adopted in the INPULSIS trial [12]. Briefly, patients who had a histologically confirmed usual interstitial pneumonia (UIP) pattern on a surgical lung biopsy (SLB) specimen were included. In the absence of a SLB specimen used for confirmation of UIP, a high-resolution computed tomography (HRCT) scan showing honeycombing and/or a combination of reticular abnormalities and traction bronchiectasis without atypical features of UIP was required for study inclusion.

Approval for the use of these data and the analysis was provided by the ethics committee of the Faculty of Medicine, Kindai University (No. 31–272). The need for informed consent was waived because of the retrospective nature of the study.

### Diagnosis of HF

The diagnosis of HF upon the hospitalization of each study patient was mainly based on the Framingham Study, as follows: 1) clinical evidence of HF such as dyspnea on exertion, paroxysmal nocturnal dyspnea or orthopnea, or peripheral edema; 2) chest radiography or HRCT revealing pulmonary venous congestion, cardiomegaly, or pleural effusion; 3) level of B-type natriuretic peptide (BNP) ≥100 pg/mL recorded at admission; 4) deterioration not explained

by other causes such as acute exacerbation of IPF, pulmonary infection, pulmonary embolism, mediastinal emphysema and/or pneumothorax, or anemia [13, 14]. The types of HF were as follows: HFpEF was defined as HF with a LVEF ≥50% and HFrEF as HF with a LVEF <50% on echocardiography. Furthermore, patients with an echocardiographic right ventricular systolic pressure (RVSP) ≥50 mm Hg were classified with HFpEF and high RVSP because the patients in this group might have had existing pulmonary hypertension (PH), including precapillary PH [15]. None of the study patients with HFrEF had a RVSP ≥50 mm Hg. Finally, we divided the IPF patients with HF into 3 groups, as follows: HFpEF without RVSP (LVEF ≥50% and RVSP <50 mm Hg), HFpEF with high RVSP (LVEF ≥50% and RVSP ≥50 mm Hg), and HFrEF (LVEF <50% and RVSP <50 mm Hg).

## Pulmonary function tests

The pulmonary function tests that were performed within 1 year prior to the hospitalization were used for baseline pulmonary function. Spirometry and single-breath measurements of diffusing capacity for carbon monoxide were performed by the CHESTAC-8800 spirometer (Chest, Tokyo, Japan), according to current international best practices [16, 17]. Results were expressed as percentages of Japanese normal predicted values [18, 19].

## Echocardiography

The LV end-systolic and end-diastolic diameters, and left atrial diameter were measured by M-mode echocardiography. LVEF was measured according to the modified Simpson method [20]. The RVSP was calculated from the estimated transtricuspid pressure gradient (TRPG) and right arterial pressure. The TRPG was estimated based on a Doppler assessment of the peak velocity of tricuspid regurgitation [15]. In our study, we used the data from the echocardiographic assessment performed at the time of admission.

## Data collection

We reviewed and recorded the clinical characteristics of all the study patients, as follows: the findings on physical examination and results of standard laboratory tests upon admission, outpatient use of long-term oxygen therapy, and IPF treatment before admission.

## Survival assessments

We evaluated the recurrence of HF and the 30- and 90-day mortality rates of the patients. Information on all the deaths was obtained from review of the hospital charts.

## Recurrence of HF

Recurrence of HF in patients with IPF was reviewed. The time-to-recurrence was defined as the number of days from the date of hospitalization for the first episode of HF during the study period until the date of hospitalization for a recurrence of HF.

## Statistical analysis

Continuous variables are presented as means ± standard deviation. Categorical variables are presented as frequencies. Comparisons between categorical variables were performed by the Fisher exact test. Comparisons between the parameters of the patients grouped according to types of HF (HFpEF without elevated RVSP, HFpEF with elevated RVSP, and HFrEF) were performed by one-way analysis of variance, followed by the Bonferroni correction for multiple comparisons. Univariate logistic regression analysis was used to identify the potential risk

factors for recurrence and 30- and 90-day mortality. For all tests, *P* < 0.05 was considered statistically significant. Analysis was performed by Statflex ver.6 software (Artech, Co., Ltd., Osaka, Osaka, Japan).

## Results

Fig 1 shows the study inclusion flowchart. Of 200 patients with IPF who were first admitted nonelectively from January 2008 through December 2018, 37 (18.5%) were hospitalized because of HF. Three were excluded because they did not undergo echocardiography. Eventually, a total of 34 patients (17.0%) were included in this study. The study patients were classified as follows: HFpEF without elevated RVSP (n = 17 [50.0%]), HFpEFwith elevated RVSP (n = 11 [32.3%]), and HFrEF (n = 6 [17.6%]). The baseline characteristics of the patients before admission are shown in Table 1. The differences between the parameters of the characteristics of the 3 study groups were not significant, except for the numbers of patients receiving corticosteroid therapy for IPF. Regarding comorbidities associated with cardiovascular disease (CVD), the rates of coronary artery disease, hypertension, diabetes mellitus, and dyslipidemia were highest in the patients with HFrEF; however, the rates of atrial fibrillation/flutter were highest in the patients with HFpEF without elevated RVSP.

The admission clinical data of patients are shown in Table 2. Significant differences were observed between the values for BNP, LV end-systolic diameter, LV end-diastolic diameter, LVEF, and RV end-systolic pressure (*P* = 0.01, *P*<0.0001, *P* = 0.0005, *P*<0.0001 and *P*<0.0001, respectively). Intergroup comparisons found that patients with HFpEF without

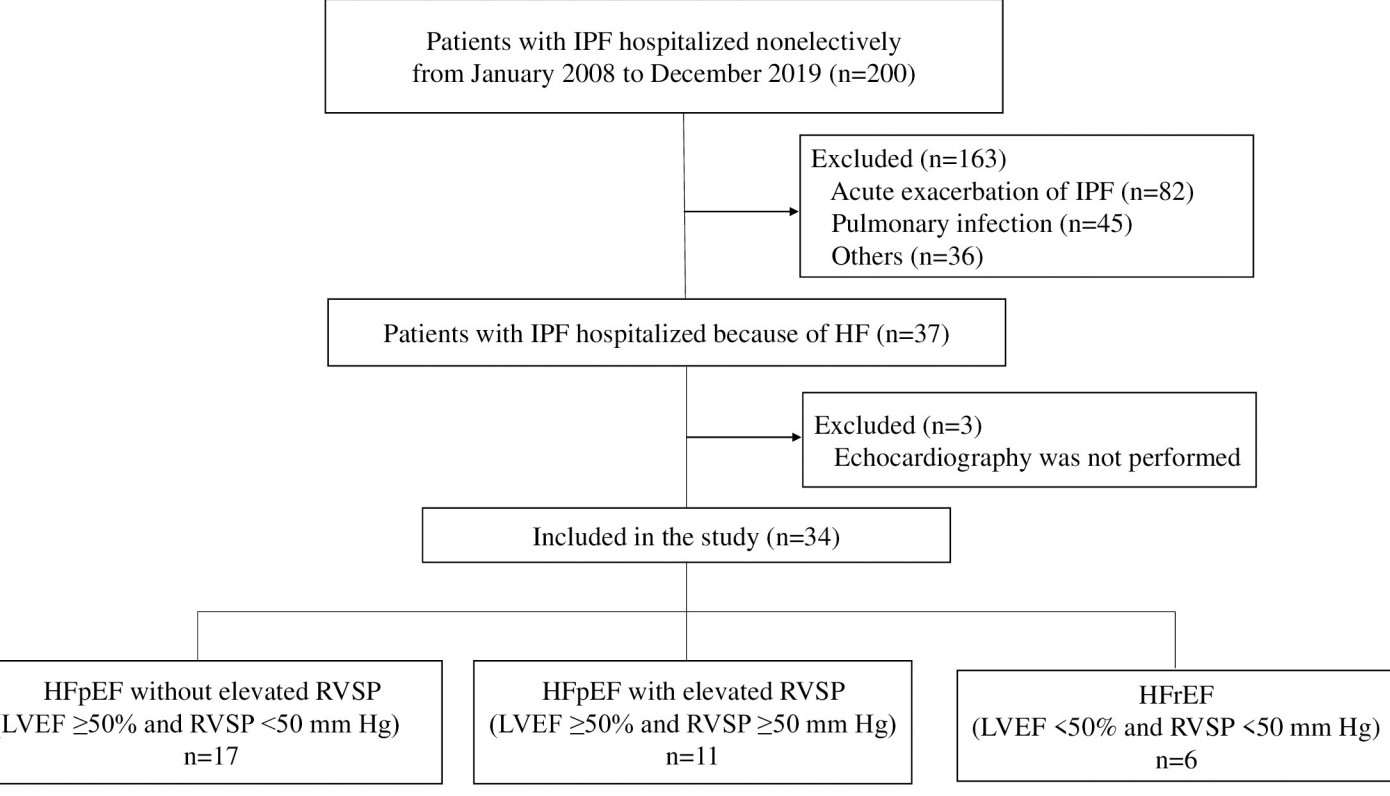

**Fig 1. Flowchart of the enrollment of study patients.** HF, heart failure; HFpEF, heart failure with preserved ejection fraction; HFrEF, heart failure with reduced ejection fraction; IPF, idiopathic pulmonary fibrosis; LVEF, left ventricular ejection fraction; RVSP, right ventricular systolic pressure.

**Table 1. Baseline characteristics and treatments of patients with IPF before admission.**

| Variables | HFpEF without elevated RVSP | HFpEF with elevated RVSP | HFrEF | |
|---|---|---|---|---|
| | **n = 17** | **n = 11** | **n = 6** | **P value** |
| Age, yr | 78.7 ± 7.3 | 75.8 ± 5.7 | 76.5 ± 6.3 | 0.51 |
| Sex, male | 12 (71) | 9 (82) | 5 (83) | 0.71 |
| Body mass index, kg/m$^2$ | 21.3 ± 4.6 | 21.4 ± 4.3 | 22.1 ± 2.4 | 0.91 |
| Pulmonary function tests | | | | |
| FVC, L | 2.0 ± 0.6[a] | 2.3 ± 0.9 [b] | 2.8 ± 0.5[c] | 0.23 |
| FVC, % predicted | 71.7 ± 20.3[a] | 72.5 ± 23.8 [b] | 80.6 ± 9.4[c] | 0.75 |
| FEV$_1$, L | 1.6 ± 0.4[a] | 1.9 ± 0.7 [b] | 2.2 ± 0.6[c] | 0.17 |
| FEV$_1$/FVC, % | 80.9 ± 10.5[a] | 85.0 ± 10.0 [b] | 77.5 ± 8.9[c] | 0.41 |
| DLco, mL/min/mmHg | 7.1 ± 1.8[d] | 6.2 ± 1.3 [e] | 5.7 ± 2.0[f] | 0.38 |
| DLco, % predicted | 55.6 ± 16.4[d] | 45.8 ± 23.0 [e] | 41.3 ± 21.8[f] | 0.43 |
| Smoking status | | | | |
| yes/no | 10 (59) | 9 (82) | 5 (83) | 0.32 |
| Long-term oxygen therapy | | | | |
| yes/no | 11 (65) | 7 (64) | 3 (50) | 0.80 |
| Treatment of IPF | | | | |
| Antifibrotic agents | 2 (12) | 4 (36) | 0 (0) | 0.11 |
| pirfenidone/nintedanib | 1/1 | 2/2 | – | |
| Corticosteroid | 7 (41) | 1* (9) | 4 (67) | 0.04 |
| Cyclosporine | 5 (29) | 0 (0) | 1 (17) | 0.13 |
| None | 9 (53) | 6 (55) | 2 (33) | 0.66 |
| Comorbidities | | | | |
| Coronary artery disease | 3 (18) | 3 (27) | 4 (67) | 0.07 |
| Hypertension | 9 (53) | 4 (36) | 5 (83) | 0.17 |
| Diabetes mellitus | 4 (24) | 2 (18) | 4 (67) | 0.08 |
| Dyslipidemia | 4 (24) | 3 (27) | 4 (67) | 0.13 |
| Atrial fibrillation/flutter | 6 (35) | 1 (9) | 0 (0) | 0.09 |
| Hypothyroidism | 0 (0) | 1 (9) | 1 (17) | 0.32 |
| Sleep apnea syndrome | 1 (6) | 0 (0) | 0 (0) | 0.59 |
| Duration of illness from diagnosis of IPF to onset of HF, years | | | | |
| | 2.4 ± 2.3 | 3.5 ± 2.5 | 3.0 ± 3.1 | 0.62 |

a: n = 12

b: n = 10

c: n = 4

d: n = 10

e: n = 3

f: n = 8

Values are shown as actual numbers or means±standard deviation. The numbers in parentheses are percentages.

* $P = 0.01$ and $P = 0.06$ compared to patients with HFpEF without high RVSP and patients with HFrEF, respectively

DLco = diffusing capacity for carbon monoxide, FEV$_1$ = forced expiratory volume in 1 second, FVC = forced vital capacity, HF = heart failure, HFpEF = heart failure with preserved ejection fraction, HFrEF = heart failure with reduced ejection fraction, IPF = idiopathic pulmonary fibrosis, PH = pulmonary hypertension, RVSP = right ventricular systolic pressure.

elevated RVSP had significantly lower BNP levels than patients with HFpEF and elevated RVSP, and patients with HFrEF ($P = 0.0004$ and $P = 0.01$, respectively). Patients with HFrEF had significantly higher values for LV end-systolic and end-diastolic diameters than patients

**Table 2. Clinical data of patients with IPF at hospitalization.**

| Variables | HFpEF without elevated RVSP | HFpEF with elevated RSVP | HFrEF | |
|---|---|---|---|---|
| | n = 17 | n = 11 | n = 6 | *P* value |
| Vital signs | | | | |
| Heart rate, /min | 82.7 ± 15.1 | 81.6 ± 18.9 | 85.5 ± 6.2 | 0.88 |
| Systolic BP, mm Hg | 126.3 ± 19.0 | 120.6 ± 18.8 | 125.8 ± 21.4 | 0.73 |
| Diastolic BP, mm Hg | 75.1 ± 18.0 | 74.6 ± 16.3 | 68.8 ± 10.6 | 0.71 |
| Laboratory data | | | | |
| Sodium, mEq/L | 136.0 ± 8.1 | 135.8 ± 8.7 | 141.6 ± 3.5 | 0.28 |
| Potassium, mEq/L | 100.2 ± 8.2 | 99.4 ± 8.8 | 106.3 ± 3.5 | 0.20 |
| Hemoglobin, g/dL | 12.8 ± 2.2 | 13.7 ± 2.7 | 12.6 ± 2.4 | 0.54 |
| BUN, mg/dL | 21.7 ± 10.5 | 17.1 ± 6.1 | 28.5 ± 9.8 | 0.06 |
| Creatinine, mg/dL | 0.9 ± 0.5 | 0.73 ± 0.2 | 1.2 ± 0.8 | 0.19 |
| BNP, pg/mL | 250 ± 146* | 733 ± 437 | 1304 ± 1687 | 0.01 |
| KL-6, U/mL | 1015 ± 611 | 754 ± 599 | 968 ± 629 | 0.53 |
| Arterial blood gas | | | | |
| pH | 7.42 ± 0.02 | 7.45 ± 0.07 | 7.44 ± 0.03 | 0.59 |
| $PaO_2 / FiO_2$ ratio | 233 ± 73 | 235 ± 89 | 211 ± 89 | 0.82 |
| $PaCO_2$ | 39.6 ± 7.4 | 40.8 ± 11.8 | 34.8 ± 6.4 | 0.41 |
| Echocardiography | | | | |
| LA diameter, mm | 38.0 ± 8.3 | 35.0 ± 7.2 | 36.0 ± 10.0 | 0.64 |
| LV systolic diameter, mm | 26.8 ± 4.2 | 26.0 ± 4.6 | 40.6 ± 5.7[†] | < 0.0001 |
| LV diastolic diameter, mm | 43.7 ± 5.6[‡] | 37.8 ± 6.3 | 51.1 ± 5.3[§] | 0.0005 |
| LVEF, % | 69.2 ± 4.7 | 68.7 ± 4.9 | 38.6 ± 8.4[‖] | <0.0001 |
| RV systolic pressure, mmHg | 38.6 ± 6.9 | 67.6 ± 12.0[¶] | 38.6 ± 10.9 | <0.0001 |

Values are shown as actual numbers or means±standard deviation

* *p* = 0.0004 and *p* = 0.01 compared to patients with HFpEF and elevated RVSP and patients with HFrEF, respectively

† *p*<0.0001 compared to patients with HFpEF with and without elevated RVSP groups

‡*P* = 0.01 compared to patients with HFpEF and elevated RVSP and patients with HFrEF

§*P* = 0.0004 compared to patients with HFpEF and elevated RVSP

‖ *P*<0.0001 compared to patients with HFpEF with and without elevated RVSP groups

¶*P*<0.0001 and *p* = 0.0002 compared to patients with HFpEF without elevated RVSP and patients with HFrEF, respectively

BNP = brain natriuretic peptide, BP = blood pressure, BUN = blood urea nitrogen, HF = heart failure, HFpEF = heart failure with preserved ejection fraction, HFrEF = heart failure with reduced ejection fraction, IPF = idiopathic pulmonary fibrosis, KL-6 = Krebs von der Lungen-6, LA = left atrium, LV = left ventricular, LVEF = left ventricular ejection fraction, $PaCO_2$ = partial pressure of carbon dioxide, $PaO_2/FiO_2$ = partial pressure of oxygen/fraction of inspiratory oxygen, PH = pulmonary hypertension, RV = right ventricular, RVSP = right ventricular systolic pressure

with HFpEF without elevated RVSP (*P*<0.0001 and *P* = 0.01, respectively) and patients with HFpEF and elevated RVSP (*P*<0.0001 and *P* = 0.0004, respectively). Patients with HFpEF without elevated RVSP had significantly higher values for LV end-diastolic diameter than patients with HFpEF and elevated RVSP (*P* = 0.01). Patients with HFrEF had significantly lower values for LVEF than patients with HFpEFs (*P*<0.0001 and *P*<0.0001, respectively). Patients with HFpEF and elevated RVSP had significantly higher values for RV end-systolic pressure than patients with HFpEF without elevated RVSP and patients with HFrEF (*P*<0.0001 and *P* = 0.0002, respectively).

The outcomes of patients during the first hospitalization are shown in Table 3. The 30-day mortality rates of patients with HFpEF without elevated RVSP, patients with HFrEF, and patients with HFpEF and elevated RSVP were 0%, 16.6%, and 36.3%, respectively, with a

**Table 3. Outcomes of patients with IPF who were hospitalized nonelectively for HF.**

| Outcome measures | HFpEF without elevated RVSP | HFpEF with elevated RVSP | HFrEF | |
|---|---|---|---|---|
| | n = 17 | n = 11 | n = 6 | P value |
| 30-day mortality, % | 0* | 36.3 | 16.6 | 0.02 |
| 90-day mortality, % | 11.7 | 45.4 | 16.6 | 0.11 |
| Recurrence rate, % | 37.5† | 12.5‡ | 16.6 | 0.29 |

Values are shown as the percentages

* P = 0.007 compared to patients with HFpEF and elevated RVSP

† n = 16 and

‡ n = 8 because patients who died during the hospitalization period were excluded.

IPF = idiopathic pulmonary fibrosis, HF = heart failure, HFpEF = heart failure with preserved ejection fraction, HFrEF = heart failure with reduced ejection fraction, PH = pulmonary hypertension, RVSP = right ventricular systolic pressure.

significantly better survival for the patients with HFpEF without elevated RVSP (P = 0.02). The 90-day mortality rates were 11.7%, 16.6%, and 45.4%, showing survival rates from best to worst for patients with HFpEF without elevated RVSP, patients with HFrEF, and patients with HFpEF and elevated RSVP, respectively; although the differences were not significant. Logistic regression analysis demonstrated that HFpEF with elevated RVSP was the only predictor of 30-day mortality (odds ratio 12.5, 95% confidential interval 1.19–125; P = 0.03). There were no significant predictors for 90-day mortality. Differences between recurrence rates for the 3 patient groups were not significant. However, the recurrence rate of patients with HFpEF without elevated RVSP was approximately 2- to 3-fold higher than the recurrence rates of the 2 other patient groups. However, no significant predictors for recurrence were identified.

## Discussion

IPF is a progressive, fibrosing interstitial pneumonia that increases in prevalence with advanced age [1]. The median age for the diagnosis of IPF is about 70 years [21, 22]. Some patients with IPF require nonelective hospitalization for HF during the clinical course of their illness [3]. We classified the patients with HF into 3 groups, those with HFpEF without elevated RVSP, HFpEF with elevated RVSP, and those with HFrEF, and compared the groups. The largest proportion of study patients had HFpEF without elevated RVSP. This group of patients showed the most favorable survival, but also showed the highest recurrence rate.

Approximately 50% of patients with HF have been reported to have a normal or nearly normal LVEF [6–8]. Old age, female sex, hypertension, elevated body mass index, smoking, and diabetes have been reported to be risk factors for HFpEF [9, 23]. Patients with HFpEF also can have a respiratory comorbidity such as chronic obstructive pulmonary disease [24, 25], which leads to increased risk of mortality [25]. Although only a few studies have reported on the relationship between interstitial lung disease and left HF, it has been suggested that some patients with IPF have HFpEF, or diastolic dysfunction [26–28]. Our study focused on LV function in patients with IPF who were hospitalized because of HF. We found that some patients with IPF had HFpEF when they were hospitalized.

The reason why patients with HFpEF without elevated RVSP recurred at the highest rate is unknown. Atrial fibrillation/flutter were most frequent in the patients with HFpEF without elevated RVSP. This may account for the highest recurrent rates of HFpEF without high RVSP among the IPF patients, because atrial fibrillation/flutter have been considered to be important risk factor of readmission for HF exacerbation [29, 30]. Some of these patients might have RV

decompensation despite not having an elevated RVSP; and the decompensated right ventricle might have led to a decreased cardiac output, which ultimately resulted in reduction of the RVSP. However, this was a retrospective study in which only a few of our study patients underwent assessment of the systolic function of their right ventricle that included measurement of the systolic excursion of the tricuspid valve annular plane. Further studies are needed to clarify the relationship between RV decompensation and the recurrence of HFpEF in IPF patients. Patients with HFpEF without elevated RVSP had a significantly higher LV end-diastolic diameter than patients with HFpEF and elevated RVSP. However, the exact reason for this finding is unclear. The finding might be accounted for by chance or be the result of a decreased volume of blood reflux into the left ventricle of those patients with HFpEF and elevated RVSP.

We defined the patients with HFpEF and elevated RVSP as those patients with an RVSP ≥50 mm Hg on echocardiography. The patients with HFpEF and elevated RVSP showed the highest mortality rate. We could not clarify whether or not the PH we identified was due to HF. Some patients with IPF and existing PH also need hospitalization. In those cases, the hospitalization may be associated with the existing PH and related right HF [10]. Patients who have IPF and chronically stable PH have a poor prognosis [31, 32]. This might account for the poor survival of our hospitalized patients with HFpEF and elevated RVSP coinciding with IPF.

Comorbidities of CVD such as hypertension, arrhythmia, diabetes mellitus, hyperlipidemia, sleep apnea syndrome, and thyroid disease are major risk factors for the progression of HF. In this study, the differences between the CVD comorbidities of the 3 study groups were not significant. However, coronary artery disease, hypertension, diabetes mellitus, and dyslipidemia were most frequent in the patients with HFrEF, but atrial fibrillation/flutter were most frequent in the patients with HFpEF without elevated RVSP. As for the relationship between CVD comorbidity and HF among IPF patients, left-sided heart disease with comorbidities of coronary artery disease or hypertension may cause HFrEF and influence its clinical course in the IPF patients with HFrEF. On the other hand, atrial fibrillation/flutter are germane to HFpEF without elevated RVSP [33]. Because atrial fibrillation/flutter is one of complications from decompensated RV of cor pulmonale [34], some IPF patients with HFpEF without elevated RVSP might be in cor pulmonale. Patients with IPF and atrial fibrillation/flutter should be evaluated for HFpEF not only at a nonelective admission but also when they are stable.

Our study has several limitations. First, the study was conducted in a retrospective fashion. Second, the study was performed at a single center with a small number of patients. Third, both the underdiagnosis and overdiagnosis of HF might have occurred, although we diagnosed HF based on the criteria proposed in the Framingham study [13]. IPF and HF share a number of common signs/symptoms such as cough, dyspnea, fatigue, and reduced exercise tolerance in patients with chronically stable disease [35–37]. In the acute setting, an accurate diagnosis is even more difficult to obtain. Fourth, HFpEF with high RVSP (possible PH) was diagnosed by echocardiography, although right heart catheterization (RHC) is required for the definitive diagnosis of PH [15]. However, performing an RHC for every patient with suspected PH who requires hospitalization because of HF is impractical, because RHC is an invasive and inconvenient procedure. When a patient is not stable, an RVSP >50 mm Hg can be used as an indirect diagnostic marker of possible PH [38, 39]. Further studies are needed that focus on hospitalized patients with IPF and PH in order to determine an accurate classification. Finally, the values for LV function and RVSP of patients before hospitalization were unknown. We could not clarify if patients with HFpEF and high RVSP at hospitalization had HF or pre-existing PH. A prospective study is needed to clarify the times of onset.

In conclusion, HFpEF is the most common type of HF that requires nonelective hospitalization in patients with IPF. Patients with HFpEF without elevated RVSP survived longer than patients with HFpEF and elevated RVSP and patients with HFrEF.

## Supporting information

**S1 Data.**

(XLSX)

## Author Contributions

**Conceptualization:** Ryo Yamazaki, Osamu Nishiyama.

**Data curation:** Ryo Yamazaki, Kazuya Yoshikawa, Sho Saeki.

**Formal analysis:** Ryo Yamazaki, Kazuya Yoshikawa, Sho Saeki.

**Investigation:** Ryo Yamazaki.

**Methodology:** Osamu Nishiyama.

**Supervision:** Osamu Nishiyama, Hiroyuki Sano, Takashi Iwanaga, Yuji Tohda.

**Validation:** Osamu Nishiyama.

**Writing – original draft:** Ryo Yamazaki.

**Writing – review & editing:** Osamu Nishiyama, Hiroyuki Sano, Takashi Iwanaga, Yuji Tohda.

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
