## [Decision Letter · Decision Letter 0]

16 Oct 2020

PONE-D-20-28865

HFpEF without high right ventricular systolic pressure  is an indicator of favorable prognosis in patients with IPF who nonelectively hospitalized with heart failure

PLOS ONE

Dear Dr. Nishiyama,

Thank you for submitting your manuscript to PLOS ONE. After careful consideration, we feel that it has merit but does not fully meet PLOS ONE’s publication criteria as it currently stands. Therefore, we invite you to submit a revised version of the manuscript that addresses the points raised during the review process.

Both reviewers found some interests in this study, but pointed out a number of criticisms that require improvement and amendment. I ask the authors to fully respond to all comments made by reviewers in the revised version.

We look forward to receiving your revised manuscript.

Kind regards,

Masataka Kuwana, MD, PhD

Academic Editor

PLOS ONE

Journal Requirements:

Reviewers' comments:

Reviewer's Responses to Questions

**Comments to the Author**

1. Is the manuscript technically sound, and do the data support the conclusions?

Reviewer #1: Partly

Reviewer #2: Yes

2. Has the statistical analysis been performed appropriately and rigorously? 

Reviewer #1: Yes

Reviewer #2: No

3. Have the authors made all data underlying the findings in their manuscript fully available?

Reviewer #1: Yes

Reviewer #2: No

4. Is the manuscript presented in an intelligible fashion and written in standard English?

Reviewer #1: Yes

Reviewer #2: Yes

5. Review Comments to the Author

Reviewer #1: General comments:

The authors demonstrated both clinical features and mortalities of the three types of HF among the IPF patients who required nonselective hospitalization. The authors demonstrated that HFpEF without high RVSP was the most common type and the lowest 30-day mortality between the nonselective hospitalization. They also demonstrated the highest recurrence rates of HFpEF without high RVSP in the present study. Their article is likely to help readers to learn the clinical features of HF among IPF patients, however, there may be several biases because the present study was conducted retrospectively in the small number of patients at a single center. Although the review for each section has been adequately addressed, several changes are required to update the manuscript.

Specific comments:

Major:

#1. Because the IPF patients shown mild restrictive disorder of pulmonary function tests and seemed to be in latent condition of lung disease before the hospitalization, cardiovascular disease (CVD) may predominantly affect their disease vulnerability in the present study. Since several CVD comorbidities, such as hypertension, arrhythmia, diabetes mellitus, hyperlipidemia, sleep apnea, and thyroid disease, possess a possibility to cause HF, the authors should describe and add the condition of above CVD comorbidities to the baseline characteristics of the IPF patients. The authors should also explain whether the patients had coronary artery disease which led to HF. Then the authors should explain the relationship between CVD and HF among the IPF patients who required nonselective hospitalization in the text.

#2. The authors demonstrated the highest 30-day mortality of HFpEF with high RVSP and speculated that the existing PH could influence the mortality among the IPF patients who required nonselective hospitalization. The authors also demonstrated the most favorable survival and the highest recurrent rates of HFpEF without high RVSP among the IPF patients. Why did HFpEF without high RVSP recur the most often among the IPF patients? In the above situations, the condition of RV should be diagnosed differentially whether decompensation or not. The authors did not demonstrate the RV systolic function, such as TAPSE, in parameters from echocardiography. If some patients were in RV decompensation despite without high RVSP, this may explain the highest recurrent rates of HFpEF without high RVSP among the IPF patients, because decompensated RV may decline cardiac output leading to reduce RVSP.

#3. According to the authors’ results, the greater values of LV end-diastolic diameters in HFpEF without high RVSP than in with high RVSP were demonstrated among the IPF patients who required nonselective hospitalization. What does it mean? Is there any pathological difference in LV morphology between HFpEF with and without high RVSP among the IPF patients?

Minor:

#1. Results, page 10: Illness duration from diagnosis of IPF to onset of HF should be demonstrated in “baseline characteristics and treatments of patients with IPF before the hospitalization”.

#2. Abstract, page 2, lines 12: “HFpEF with a high RVSP” should be changed to “HFpEF with high RVSP”.

#3. Introduction, page 4, line 16: “,” should be added following “HF with reduced EF (HFrEF)”.

#4. Results, page 9, line 9: “HF” should be changed to “HFpEF”.

#5. Results, page 11, line 4: What does each “n” mean? The number of each “n” seems to be greater than the number of each population in the present study.

#6. Results, page 12, lines 10-11: “a significantly higher value” should be changed to “significantly higher values”.

#7. Discussion, page 18, line 6: “chronic, stable” should be changed to “chronically stable”.

#8. Figure legend, page 27, line 5: “PH, pulmonary hypertension” should be removed because the figure does not include the terms.

Reviewer #2: The author investigated the clinical features and outcomes in patients with IPF and HF who required nonelective hospitalization. They demonstrated that among patients with IPF, HFpEF with RVSP elevation had worth outcome compared to other forms of HF.

Though this study will draw attention from substantial numbers of experts, I found some concerns on the study that needed to be revised.

1. The author demonstrated table 1 as baseline characteristics and treatment with IPF before the hospitalization. This table could make readers some confusion.

The patients background such as age, BMI, oxygen therapy, smoking status, and treatment should be the one on or just before admission.

So, the title should be changed from ‘before the hospitalization’ to ‘on admission’. As author mentioned in the manuscript that pulmonary function test was performed within one-year prior to the hospitalization, I think it is still acceptable to provide the data in the table.

2. The majority in cause of HFpEF is left diastolic dysfunction. So, the author should provide parameters associated with diastolic dysfunction such as E/e’, and LAVI in addition to LAD and LVD by echocardiography if available.

3. Since diabetes, hypertension, and arrythmia are kwon to associated with diastolic dysfunction, these data should be provided in the background of the patients in the manuscript.

4. Was the mortality analyzed using Kaplan-Meier? It should be clearly written in a manuscript. And the survival and recurrence rate are preferably provided with a figure.

5. The independent risk factor for mortality and recurrence need to be analyzed to show that high value of RVSP is independently associated with mortality or recurrence.

6. PLOS authors have the option to publish the peer review history of their article (what does this mean?). If published, this will include your full peer review and any attached files.

Reviewer #1: No

Reviewer #2: No

---

## [Author Response · Author response to Decision Letter 0]

16 Dec 2020

Dear Dr. Masataka Kuwana,

Thank you very much for reviewing our submitted manuscript. We hope that we have addressed all of the reviewers' concerns to your satisfaction. 

We also revised some expressions including the title according to a native speaker’s correction.

To Reviewer 1

For the major comments:

1. Because the IPF patients shown mild restrictive disorder of pulmonary function tests and seemed to be in latent condition of lung disease before the hospitalization, cardiovascular disease (CVD) may predominantly affect their disease vulnerability in the present study. Since several CVD comorbidities, such as hypertension, arrhythmia, diabetes mellitus, hyperlipidemia, sleep apnea, and thyroid disease, possess a possibility to cause HF, the authors should describe and add the condition of above CVD comorbidities to the baseline characteristics of the IPF patients. The authors should also explain whether the patients had coronary artery disease which led to HF. Then the authors should explain the relationship between CVD and HF among the IPF patients who required nonselective hospitalization in the text.

Response: We thank Reviewer 1 for this valuable comment. According to Reviewer 1’s suggestion, we added information on the frequencies of cardiovascular disease (CVD) comorbidities, including coronary artery disease, to Table 1. We showed the number of patients with frequencies in parentheses, and therefore changed some of the format of Table 1. We added the following sentences to the Results (P9, L14) and the Discussion section (P19, L6): “Regarding comorbidities associated with cardiovascular disease (CVD), the rates of coronary artery disease, hypertension, diabetes mellitus, and dyslipidemia were highest in the patients with HFrEF; however, the rates of atrial fibrillation/flutter were highest in the patients with HFpEF without elevated RVSP.”, and “Comorbidities of CVD such as hypertension, arrhythmia, diabetes mellitus, hyperlipidemia, sleep apnea syndrome, and thyroid disease are major risk factors for the progression of HF. In this study, the differences between the CVD comorbidities of the 3 study groups were not significant. However, coronary artery disease, hypertension, diabetes mellitus, and dyslipidemia were most frequent in the patients with HFrEF, but atrial fibrillation/flutter were most frequent in the patients with HFpEF without elevated RVSP. Patients with IPF and atrial fibrillation/flutter should be evaluated for HFpEF not only at a nonelective admission but also when they are stable.”

2. The authors demonstrated the highest 30-day mortality of HFpEF with high RVSP and speculated that the existing PH could influence the mortality among the IPF patients who required nonselective hospitalization. The authors also demonstrated the most favorable survival and the highest recurrent rates of HFpEF without high RVSP among the IPF patients. Why did HFpEF without high RVSP recur the most often among the IPF patients? In the above situations, the condition of RV should be diagnosed differentially whether decompensation or not. The authors did not demonstrate the RV systolic function, such as TAPSE, in parameters from echocardiography. If some patients were in RV decompensation despite without high RVSP, this may explain the highest recurrent rates of HFpEF without high RVSP among the IPF patients, because decompensated RV may decline cardiac output leading to reduce RVSP.

Response: I understand Reviewer 1’s concern. Unfortunately, the RV systolic function was not adequately evaluated because of the retrospective nature of this study. However, we added the following sentences to comment on this issue (P18, L3). “The reason why patients with HFpEF without elevated RVSP recurred at the highest rate is unknown. Some of these patients might have RV decompensation despite not having an elevated RVSP; and the decompensated right ventricle might have led to a decreased cardiac output, which ultimately resulted in reduction of the RVSP. However, this was a retrospective study in which only a few of our study patients underwent assessment of the systolic function of their right ventricle that included measurement of the systolic excursion of the tricuspid valve annular plane. Further studies are needed to clarify the relationship between RV decompensation and the recurrence of HFpEF in IPF patients.”

3. According to the authors’ results, the greater values of LV end-diastolic diameters in HFpEF without high RVSP than in with high RVSP were demonstrated among the IPF patients who required nonselective hospitalization. What does it mean? Is there any pathological difference in LV morphology between HFpEF with and without high RVSP among the IPF patients?

Response: Thank you for your important comments. We added the following sentences to comment on this issue in the Discussion section (P18, L11). “Patients with HFpEF without elevated RVSP had a significantly higher LV end-diastolic diameter than patients with HFpEF and elevated RVSP. However, the exact reason for this finding is unclear. The finding might be accounted for by chance or be the result of a decreased volume of blood reflux into the left ventricle of those patients with HFpEF and elevated RVSP. ”

For the minor comments:

1. Results, page 10: Illness duration from diagnosis of IPF to onset of HF should be demonstrated in “baseline characteristics and treatments of patients with IPF before the hospitalization”.

Response: According to the reviewer’s suggestion, we added information on the duration of illness from the diagnosis of IPF to the onset of HF to Table 1.

2. Abstract, page 2, lines 12: “HFpEF with a high RVSP” should be changed to “HFpEF with high RVSP”.

Response: Changed as suggested, but to “HFpEF and elevated RVSP”. 

3. Introduction, page 4, line 16: “,” should be added following “HF with reduced EF (HFrEF)”.

Response: Changed as suggested. 

4. Results, page 9, line 9: “HF” should be changed to “HFpEF”.

Response: Thank you for your comments. We made the correction as suggested. 

5. Results, page 11, line 4: What does each “n” mean? The number of each “n” seems to be greater than the number of each population in the present study.

Response: Thank you for your comments. This was our mistake. We revised Table 1.

6. Results, page 12, lines 10-11: “a significantly higher value” should be changed to “significantly higher values”.

Response: Changed as suggested. 

7. Discussion, page 18, line 6: “chronic, stable” should be changed to “chronically stable”.

Response: Changed as suggested. 

8. Figure legend, page 27, line 5: “PH, pulmonary hypertension” should be removed because the figure does not include the terms.

Response: Changed as suggested. 

To Reviewer 2

1. The author demonstrated table 1 as baseline characteristics and treatment with IPF before the hospitalization. This table could make readers some confusion.

The patients background such as age, BMI, oxygen therapy, smoking status, and treatment should be the one on or just before admission.

So, the title should be changed from ‘before the hospitalization’ to ‘on admission’. As author mentioned in the manuscript that pulmonary function test was performed within one-year prior to the hospitalization, I think it is still acceptable to provide the data in the table.

Response: Thank you for your valuable comments. We changed Table 1 as suggested.

2. The majority in cause of HFpEF is left diastolic dysfunction. So, the author should provide parameters associated with diastolic dysfunction such as E/e’, and LAVI in addition to LAD and LVD by echocardiography if available.

Response: I understand Reviewer 2’s concern. Unfortunately, RV systolic function was not adequately evaluated because of the retrospective nature of this study, which was also our response to Reviewer 1. However, we added the following sentences to comment on this (P18, L3). “The reason why patients with HFpEF without high RVSP recurred at the highest rate is unknown. Some of these patients might have RV decompensation despite not having an elevated RVSP; and the decompensated right ventricle might have led to a decreased cardiac output, which ultimately resulted in reduction of the RVSP. However, this was a retrospective study in which only a few of our study patients underwent assessment of the systolic function of their right ventricle that included measurement of the systolic excursion of the tricuspid valve annular plane. Further studies are needed to clarify the relationship between RV decompensation and the recurrence of HFpEF in IPF patients.”

3. Since diabetes, hypertension, and arrythmia are kwon to associated with diastolic dysfunction, these data should be provided in the baiockground of the patients in the manuscript.

Response: We thank Reviewer 2 for this valuable comment. According to this suggestion, we added the information on the frequency of cardiovascular disease comorbidities to Table 1. 

4. Was the mortality analyzed using Kaplan-Meier? It should be clearly written in a manuscript. And the survival and recurrence rate are preferably provided with a figure.

Response: Thank you for your valuable suggestion. We calculated the 30-d and 90-d mortalities using the actual numbers of deceased patients. The 30-d and 90-d mortalities were relatively short-term mortalities. Therefore, Kaplan-Meier survival analysis with the resulting curves is not suitable. We would greatly appreciate it if Reviewer 2 would accept our decision not to include Kaplan-Meier survival curves. However, to clarify the issue, we added the following sentence to the Methods section (P8, L17). “Univariate logistic regression analysis was used to identify the potential risk factors for recurrence and 30- and 90-day mortality.” In addition, we added the following sentences to the Results section (P15, L8 and L14). “Logistic regression analysis demonstrated that HFpEF with elevated RVSP was the only predictor of 30-day mortality (odds ratio 12.5, 95% confidential interval 1.19-125; P = 0.03). There were no significant predictors for 90-day mortality.” and “However, no significant predictors for recurrence were identified.”

5. The independent risk factor for mortality and recurrence need to be analyzed to show that high value of RVSP is independently associated with mortality or recurrence.

Response: We thank Reviewer 2 for this valuable comment. However, as mentioned in our response to your previous comment, no other variable than HFpEF with elevated RVSP was predictable for survival and recurrence. We added an explanation to make it clear as responded to the previous comment.

---

## [Decision Letter · Decision Letter 1]

29 Dec 2020

PONE-D-20-28865R1

HFpEF without elevated right ventricular systolic pressure is a favorable prognostic indicator in patients with IPF requiring hospitalization for heart failure

PLOS ONE

Dear Dr. Nishiyama,

Thank you for submitting your manuscript to PLOS ONE. After careful consideration, we feel that it has merit but does not fully meet PLOS ONE’s publication criteria as it currently stands. Therefore, we invite you to submit a revised version of the manuscript that addresses the points raised during the review process.

This manuscript was improved by revision in some extent, but still remains unsatisfactory for one of the reviewers. I ask the authors to respond to all criticisms to further improve the quality.

We look forward to receiving your revised manuscript.

Kind regards,

Masataka Kuwana, MD, PhD

Academic Editor

PLOS ONE

Reviewers' comments:

Reviewer's Responses to Questions

**Comments to the Author**

1. If the authors have adequately addressed your comments raised in a previous round of review and you feel that this manuscript is now acceptable for publication, you may indicate that here to bypass the “Comments to the Author” section, enter your conflict of interest statement in the “Confidential to Editor” section, and submit your "Accept" recommendation.

Reviewer #1: All comments have been addressed

Reviewer #2: All comments have been addressed

2. Is the manuscript technically sound, and do the data support the conclusions?

Reviewer #1: Partly

Reviewer #2: Yes

3. Has the statistical analysis been performed appropriately and rigorously? 

Reviewer #1: Yes

Reviewer #2: Yes

4. Have the authors made all data underlying the findings in their manuscript fully available?

Reviewer #1: Yes

Reviewer #2: Yes

5. Is the manuscript presented in an intelligible fashion and written in standard English?

Reviewer #1: Yes

Reviewer #2: Yes

6. Review Comments to the Author

Reviewer #1: General comments:

The authors have added new data and updated the discussion in the revised version of their manuscript. Thanks for their clear response for each review. However, their speculation and interpretation are remaining a little poor in the text. Although the review for each section has been adequately addressed, several changes are required to update the manuscript.

Specific comments:

Major:

#1. According to the authors’ new data, coronary artery disease (CAD), hypertension (HTN), diabetes mellitus, and dyslipidemia were predominant in HFrEF, while atrial fibrillation/flutter was predominant in HFpEF without elevated RVSP among the IPF patients who required nonelective hospitalization. There may be clinical differences between HFrEF and HFpEF without elevated RVSP among the IPF patients. Left-sided heart disease (LHD) with comorbidities of CAD or HTN may influence its clinical course in the IPF patients with HFrEF whereas LHD may not necessarily do in the IPF patients with HFpEF without elevated RVSP. The authors did not clearly answer the comment “the authors should explain the relationship between CVD and HF among the IPF patients who required nonelective hospitalization in the text”.

#2. Since the authors demonstrated the predominance of atrial fibrillation/flutter in the IPF patients with HFpEF without elevated RVSP, the result should be explained and interpreted in the text. Because atrial fibrillation/flutter is one of complications from decompensated RV of cor pulmonale, the authors should mention this if some IPF patients with HFpEF without elevated RVSP were in cor pulmonale. The authors should also speculate a possibility that atrial fibrillation/flutter may account for the highest recurrent rates of HFpEF without high RVSP among the IPF patients.

Minor:

#1. Results, Table 1, page 10: The title “…. on admission” may be changed to “…. before admission”.

#2. Discussion, page 19, line 4: “…. our hospitalized study patients” may be changed to “…. our hospitalized patients”.

Reviewer #2: The manuscript is now much improved according to the suggestions.

I have no further comments to authors.

7. PLOS authors have the option to publish the peer review history of their article (what does this mean?). If published, this will include your full peer review and any attached files.

Reviewer #1: No

Reviewer #2: No

---

## [Author Response · Author response to Decision Letter 1]

2 Jan 2021

Dear Dr. Masataka Kuwana

Thank you kindly for reviewing our revised manuscript. We hope we have addressed all of the reviewer's concerns to your satisfaction this time. We also revised the order of the references because some were not in correct order in the previous version.

To Reviewer 1

For the major comments:

1. According to the authors’ new data, coronary artery disease (CAD), hypertension (HTN), diabetes mellitus, and dyslipidemia were predominant in HFrEF, while atrial fibrillation/flutter was predominant in HFpEF without elevated RVSP among the IPF patients who required nonelective hospitalization. There may be clinical differences between HFrEF and HFpEF without elevated RVSP among the IPF patients. Left-sided heart disease (LHD) with comorbidities of CAD or HTN may influence its clinical course in the IPF patients with HFrEF whereas LHD may not necessarily do in the IPF patients with HFpEF without elevated RVSP. The authors did not clearly answer the comment “the authors should explain the relationship between CVD and HF among the IPF patients who required nonelective hospitalization in the text”.

Response: Thank you for your important comments and valuable suggestions. To comment on the relationship between CVD and HF among the IPF patients who required nonelective hospitalization, we added followings in Discussion section (P19,L15): “As for the relationship between CVD comorbidity and HF among IPF patients, left-sided heart disease with comorbidities of coronary artery disease or hypertension may cause HFrEF and influence its clinical course in the IPF patients with HFrEF. On the other hand, atrial fibrillation/flutter are germane to HFpEF without elevated RVSP.”.

2. Since the authors demonstrated the predominance of atrial fibrillation/flutter in the IPF patients with HFpEF without elevated RVSP, the result should be explained and interpreted in the text. Because atrial fibrillation/flutter is one of complications from decompensated RV of cor pulmonale, the authors should mention this if some IPF patients with HFpEF without elevated RVSP were in cor pulmonale. The authors should also speculate a possibility that atrial fibrillation/flutter may account for the highest recurrent rates of HFpEF without high RVSP among the IPF patients.

Response: Thank you for your valuable comments. To respond to the reviewer’s comment, we added followings in Discussion section (P19,L18): “Because atrial fibrillation/flutter is one of complications from decompensated RV of cor pulmonale, some IPF patients with HFpEF without elevated RVSP might be in cor pulmonale.”. We also added followings (P18,L4): “Atrial fibrillation/flutter were most frequent in the patients with HFpEF without elevated RVSP. This may account for the highest recurrent rates of HFpEF without high RVSP among the IPF patients.”.

For the minor comments:

1. Results, Table 1, page 10: The title “…. on admission” may be changed to “…. before admission”.

Changed as suggested.

2. Discussion, page 19, line 4: “…. our hospitalized study patients” may be changed to “…. our hospitalized patients”.

Changed as suggested.

---

## [Decision Letter · Decision Letter 2]

5 Jan 2021

PONE-D-20-28865R2

HFpEF without elevated right ventricular systolic pressure is a favorable prognostic indicator in patients with IPF requiring hospitalization for heart failure

PLOS ONE

Dear Dr. Nishiyama,

Thank you for submitting your manuscript to PLOS ONE. After careful consideration, we feel that it has merit but does not fully meet PLOS ONE’s publication criteria as it currently stands. Therefore, we invite you to submit a revised version of the manuscript that addresses the points raised during the review process.

The manuscript has been improved by two rounds of revisions, but there still remains minor points requiring modifications. 

We look forward to receiving your revised manuscript.

Kind regards,

Masataka Kuwana, MD, PhD

Academic Editor

PLOS ONE

Reviewers' comments:

Reviewer's Responses to Questions

**Comments to the Author**

1. If the authors have adequately addressed your comments raised in a previous round of review and you feel that this manuscript is now acceptable for publication, you may indicate that here to bypass the “Comments to the Author” section, enter your conflict of interest statement in the “Confidential to Editor” section, and submit your "Accept" recommendation.

Reviewer #1: All comments have been addressed

2. Is the manuscript technically sound, and do the data support the conclusions?

Reviewer #1: Yes

3. Has the statistical analysis been performed appropriately and rigorously? 

Reviewer #1: Yes

4. Have the authors made all data underlying the findings in their manuscript fully available?

Reviewer #1: Yes

5. Is the manuscript presented in an intelligible fashion and written in standard English?

Reviewer #1: Yes

6. Review Comments to the Author

Reviewer #1: General comments:

The authors have updated the text in the revised version of their manuscript. Thanks for their clear response for each review. Almost concerns have been adequately addressed in the manuscript.

Minor:

#1. The authors have added new speculations and updated the discussion in the revised version of their manuscript. However, their speculations are without quoting of any references. The authors should describe their speculations based on references.

#2. Discussion, page 20, line 1: “chronic stable” should be changed to “chronically stable”.

7. PLOS authors have the option to publish the peer review history of their article (what does this mean?). If published, this will include your full peer review and any attached files.

Reviewer #1: No

---

## [Author Response · Author response to Decision Letter 2]

7 Jan 2021

Dear Dr. Masataka Kuwana

Thank you kindly for reviewing our submitted revised manuscript. We again tried to address the reviewers' concerns. 

To Reviewer 1

For the comments:

1. The authors have added new speculations and updated the discussion in the revised version of their manuscript. However, their speculations are without quoting of any references. The authors should describe their speculations based on references.

Response: We thank the reviewer for valuable suggestion. We added following references with a short explanation in Discussion section.

29. Koitabashi T, Inomata T, Niwano S, Nishii M, Takeuchi I, Nakano H, et al. Paroxysmal atrial fibrillation coincident with cardiac decompensation is a predictor of poor prognosis in chronic heart failure. Circ J. 2005;69:823-830. 

30. Ahmed MI, White M, Ekundayo OJ, Love TE, Aban I, Liu B, et al. A history of atrial fibrillation and outcomes in chronic advanced systolic heart failure: a propensity-matched study. Eur Heart J. 2009;30:2029-2037.

33. Reddy YNV, Obokata M, Gersh BJ, Borlaug BA. High prevalence of occult heart failure with preserved ejection fraction among patients with atrial fibrillation and dyspnea. Circulation. 2018;137:534-535.

34. Obokata M, Reddy YN, Melenovsky V, Pislaru S, Borlaug BA. Deterioration in right ventricular structure and function over time in patients with heart failure and preserved ejection fraction. Eur Heart J 2019;40:689-697.

2. Discussion, page 20, line 1: “chronic stable” should be changed to “chronically stable”.

Response: Changed as suggested.

---

## [Decision Letter · Decision Letter 3]

8 Jan 2021

HFpEF without elevated right ventricular systolic pressure is a favorable prognostic indicator in patients with IPF requiring hospitalization for heart failure

PONE-D-20-28865R3

Dear Dr. Nishiyama,

We’re pleased to inform you that your manuscript has been judged scientifically suitable for publication and will be formally accepted for publication once it meets all outstanding technical requirements.

Kind regards,

Masataka Kuwana, MD, PhD

Academic Editor

PLOS ONE

Additional Editor Comments (optional):

Reviewers' comments:

Reviewer's Responses to Questions

**Comments to the Author**

1. If the authors have adequately addressed your comments raised in a previous round of review and you feel that this manuscript is now acceptable for publication, you may indicate that here to bypass the “Comments to the Author” section, enter your conflict of interest statement in the “Confidential to Editor” section, and submit your "Accept" recommendation.

Reviewer #1: All comments have been addressed

2. Is the manuscript technically sound, and do the data support the conclusions?

Reviewer #1: Yes

3. Has the statistical analysis been performed appropriately and rigorously? 

Reviewer #1: Yes

4. Have the authors made all data underlying the findings in their manuscript fully available?

Reviewer #1: Yes

5. Is the manuscript presented in an intelligible fashion and written in standard English?

Reviewer #1: Yes

6. Review Comments to the Author

Reviewer #1: General comments:

The authors have updated the text in the revised version of their manuscript. Thanks for their clear response for each review. Almost concerns have been adequately addressed in the manuscript. The reviewer will be without any regret for the revised version of their manuscript.

7. PLOS authors have the option to publish the peer review history of their article (what does this mean?). If published, this will include your full peer review and any attached files.

Reviewer #1: No

---

## [Editor Report · Acceptance letter]

12 Jan 2021

PONE-D-20-28865R3 

HFpEF without elevated right ventricular systolic pressure is a favorable prognostic indicator in patients with IPF requiring hospitalization for heart failure 

Dear Dr. Nishiyama:

I'm pleased to inform you that your manuscript has been deemed suitable for publication in PLOS ONE. Congratulations! Your manuscript is now with our production department. 

Kind regards, 

on behalf of

Prof. Masataka Kuwana 

Academic Editor

PLOS ONE